

# Exploring Geomorphic Processes and Martian Gale Crater Topography on Mars using CTX and HiRISE Express Image Dataset

Pavan Kumar[1], Jiwan Singh Rawat[1], Sufia Rehman[2], Haroon Sajjad[2], Meenu Rani[1], Kirti Kumar Mohanto[1], Osheen Rai[1], Bismay Tripathy[1]

[1]Department of Remote Sensing and GIS, Kumaun University, SSJ Campus, Almora-263601, Uttarakhand, India
[2]Department of Geography, Facult of Natural Sciences, Jamia Millia Islamia, New Delhi-110005, India

*Correspondence to*: Meenu Rani (meenurani06@gmail.com)

**Abstract.** Exploration of Martian surface and the crater deposition has recently attracted scientific community. We hypothesized the existence of momentous topographic features of different origin on the Martian surface. It was observed that Gale Crater has a thick sediment deposition mainly of fluvial and aeolian origin. This study has utilized spatially referenced topographic dataset Context Camera images acquired from Mars reconnaissance orbiter for exploring the geomorphic processes and topography of Gale Crater. A base map was prepared by mosaicking all images for preparing geomorphologic map of the crater. Surface map of the topography of the Crater was prepared using Mars Orbiter Laser Altimeter (MOLA) data. HiRISE images were used to examine the identified geomorphic features. Results revealed that the presence of both fluvial and aeolian processes and their respective associated landforms in the Crater. Depositional landforms such as alluvial fan, inverted channel and mound and erosional landform namely canyons were distinctly identified on the image. Yardangs produced by erosional and ripple and dunes formed by the depositional work of wind were also identified in the Crater. The fluvial channel, inverted channels and fan shaped deposits further signifies ongoing aqueous activity on the Gale Crater. Sinuous ridges are the common features present on the floor of Crater. Rock particles, sand and silt in the crater were found to have been transported and deposited by flowing water on its floor. Presence of different sedimentary structures and valley revealed ancient sedimentary deposition due to water action. Significant ejecta morphologies were also identified on Martian surface. Our study confirms the previous studies that presence of $H_2O$ sub surface volatiles. Concentrations of volatiles have produced double layer ejecta morphology. The concept and findings of this study will escalate knowledge about the surface features on Gale Crater. However, more coherent investigation is needed for modelling and understanding the processes and landforms of the crater. Context Camera (CTX) and High-Resolution Imaging Science Experiment (HiRISE) images have proved useful for geomorphic and topographic mapping of the Mars planet. The dataset used in this study can be accessed on Mars Orbital Data Explorer (https://ode.rsl.wustl.edu/mars/indexproductsearch.aspx).



# 1 Introduction

Studies on extra-terrestrial surfaces and associated geological processes allured scientific community worldwide. One such important attempt is associated with analyzing the martian surface topography and related processes. Impact craters are the most obvious topographic features on the non-gaseous bodies of the solar system which have been originated through constant natural fusillade. Gale Crater with a diameter of nearly 154 meters is believed to be evolved between 3.5 and 3.8 billion years ago. Mars Science Laboratory Curiosity Rovers (MSLCR) identified Gale Crater as one of the prominent landing sites for exploring geomorphology (Golombek *et al.,* 2009; Golombek *et al.,* 2003). The crater is composed of thick sediment deposition, bowl shaped depression and a central mountain. The mountain in the crater is completely enclosed by walls. Studies conducted on origin of Gale Crater revealed contrasting opinion among scientists around the world. Several scholars have demonstrated that geomorphic features have been produced by mass wasting, pyroclastic sedimentation, fluvial, aeolian and lava deposition on the surface of Mars (Carl and Bagnold, 1975; Scott *et al.* 1978; Greeley and Guest, 1987; Scott and Chapman, 1995). Mound and valleys in the crater supported the existence of aqueous processes (Pelkey and Jakosky, 2002). Scientists working on dating of rocks suggested the formation of rocks between 4560 and 3860 million years ago and these surfaced between 48 and 108 million years ago. Clastic volcanic sediments, aeolian and marine processes have contributed deposition of thick sediments on Mars surface. Most of the features on Gale crater belonged to lacustrine sediments (Malin and Edgett, 2000). Impact and volcanic craters cover most of Martian surface. These craters were originated about 3 billion years ago and have undergone through various geological processes mainly erosion and deposition (Hartmann *et al.,* 2001). Grant et al. (2014) identified significant mound on the Gale Crater namely Mt. Sharp and Aeolis Mons which were expected to evolve between late Hesperian and early Amazonian periods. They examined the crater material by integrating the geological maps and the crater statistics. The scholars also signified existence of mound deposition due to impact crater and aeolian and fluvial processes. Recently, Mars Odyssey, Mars Reconnaissance Orbiter (MRO) and Mars Global Surveyor (MGS) have been utilized to evaluate the Martian impact crater. These space craft's have provided a large volume of multispectral, stereo and topographic data (Albee *et al*., 1992; Chapman *et al*., 2003). Mars is stratified into Amazonian, Hesperian and Noachian time scales on the basis of the crater size distribution per square kilometer (Tanaka, 1986). The crater is believed to be formed during Noachian age (Bridges, 2001; Cabrol *et al*., 1999).

Presence of valley's sedimentary structures indicated the existence of ancient sedimentary deposits and work of water in the crater. Aeolian features including yardang are also present in the crater. Researchers further identified crater partially filled with water (Malin and Edgett, 2000) and erosion on the mound due to subsequent erosion and deposition. Pelkey *et al*. (2004) evaluated the role of aeolian processes in shaping the existent mound and crater. The study also examined the role of aqueous processes in formation of valley floor of Gale Crater. No agreement on the single process was made by earlier studies. For example, Rossi *et al*. (2008) ruled out the possibility of fluvial activity around the crater while some other scholars have reported fan shaped mesas and ridges existed along inverted fluvial channels and alluvial fans (Thomson *et al*.





2008). Later some authors noticed that upper mound material belonging to Medusae Fossae Formation (MFF) which further transformed into yardang (Zimbelman *et al.*, 2010). Cadieux (2011) suggested the growth of atmospherically transported sediment in the form of ash, dust, impact ejecta, ice nuclei, sand, etc. had created sedimentary deposits and supported aeolian process. Desai and Murty (2016) advocated the existence of fluvial and aeolian activities on mars using morphological analysis and high-resolution Context Camera (CTX) dataset (Scholten *et al.*, 2005; Murchie *et al.*, 2007; Gwinner *et al.*, 2009; Bell III *et al.*, 2013). Martian topography includes layers of ejecta material namely single, double, multiple and radial ejecta layers (Weiss and Head, 2014).

Anderson and Bell (2010) analysed the topographic features on martian surface using Reconnaissance Orbiter Context Camera radiometrically calibrated data as the base map. They used high resolution imaging science experiment (HiRISE) and thermal infrared datasets to examine the stereographic origin of the topographic features and the ancillary physical characteristics of martian surface. They identified that diverse nature of the topographic features owe their origin from both aeolian and fluvial processes. de Villiers *et al.* (2011) advocated that high gradient alluvial fans on the martian surface supported the dominance of debris flow processes. Aharonson *et al.* (2001) analyzed the slope, statistical characteristics of geomorphic features and future landing sites on Mars using Mars Orbiter Laser Altimeter (MOLA) datasets. Slope variation was differentiated using three-point Lagrange formula. They suggested that the northern morphological structure was formed earlier than the formation of volcanic southern structure. Le Deit *et al.* (2013) explored the formation of sediments of Aeolis Mons on Gale Crater by analyzing the age of crater's ejecta. They identified colluvial sediment deposition on the wall of Aeolis Mons where northern portion was having significant mass wasting deposition. In view of diversified findings related to the origin of the geomorphic features on the surface of Mars, present study intends to envisage the process of evolution of present form of crater and resultant geomorphological features. The study credibly utilized visible and topographic datasets obtained from CTX, HiRISE images for mapping and interpreting the geomorphic features on the Gale Crater. Findings of the study may excel in augmenting the further exploration of Gale Crater and other landing sites.

**2 Study area**

Gale Crater (5.3°S, 222.3°W) with 170 km in diameter lies between the southern cratered highlands and Elysium Planitia low lands in north-eastern part of the Aeolis quadrangle. The inset vertical exaggeration image provides 3D view of the Gale Crater and its surrounding area (Fig. 1). Within the crater, an off-centred mound is present slightly towards the southern side. The lower most section of the mound showed an upward transition of formation from clay-bearing to sulphate-bearing beds (Milliken *et al.,* 2010). The topographically mound layer is higher than the northern rim and gives an appearance of high mound. However, some portion of the southern rim is slightly higher than the mound. The mound height indicated that entire crater was once filled with sediment. The place between the northern rim and the foot hill of Aeolis Mons is named as Aeolis Palus where the mars curiosity rover was landed by National Aeronautics and Space Administration (NASA).

<Insert Figure 1. Mars Orbiter Laser Altimeter (MOLA) global topographic map of mars and inset map showing the vertical exaggeration view of Gale Crater and its surrounding >

## 3 Material and methodology

### 3.1 Datasets used for morphological analysis

The topographic dataset of context camera (CTX) images with 6 meters spatial resolution obtained from the Mars Reconnaissance Orbiter satellite were used for mapping the Martian geomorphological features. The detailed methodology for mapping and identification of geomorphic features is presented in Fig. 2. Base map was prepared by mosaicking all the acquired CTX images with the help of Arc GIS software. Mars Orbiter Laser Altimeter (MOLA) data was acquired by Mars Global Surveyor spacecraft. The satellite data obtained from Mars orbital data explorer were used for delineating the topographic features of the surface of Gale Crater. HiRISE (~0.25 m/pixel) dataset was used for examining the stereographic characteristics of geomorphic features. Information of image used for mapping geomorphic features is presented in Table 1 and 2.

<Insert Table 1 Indent of CTX images used for fluvial and aeolian process landforms of canyon, alluvial fan, inverted channel, mound, ripples, dunes and yardangs >

<Insert Figure 2. Flowchart showing methodology for mapping and identification of geomorphological features >

<Insert Table 2 HiRISE images identity used for fluvial and aeolian process landforms of canyon, alluvial fan, inverted channel, mound, ripples, dunes and yardangs >

### 3.2 Crater morphology identification

Endeavouring the past geological upliftment and transformation of each stratum is the major concern of the scientific community. Gale crater has experienced multi-stage erosional processes. The northern rim has been partially eroded and led to formations of canyons. Mass wasted material produced by some portion of rim resulted in the deposition of fine sediments covering the floor. Geomorphological features of fluvial and aeolian processes in and around the crater were identified with the help of DN values and HiRISE images (Fig. 3).

### 3.3 Pre-processing of satellite data

Pre-processing was carried out to escalate the interpretability of the datasets. Radiometric and geometric distortions of images were removed using mathematical transformation during the image pre-processing stage. Geometric correction was



performed by employing transformation, $T$, and a distorted image, $f(x, y)$, was converted into a corrected image, $f(x', y')$, namely, $f(x', y')$ equal to $f(x, y)\, T$ for all $x, y, x'\, and\, y'$. This correction encompassed system geometry, empirical functions, scale changes and variable map projection. Projection transformation was approximated on a sub image basis by linear transformation. This transformation, $T$, is expressed as:

$$T = \begin{vmatrix} a_{11} & a_{12} & a_{13} \\ a_{21} & a_{22} & a_{23} \\ a_{31} & a_{32} & a_{33} \end{vmatrix} for\ the\ subimage, \qquad\text{(i)}$$

and the corrected grid position, $x', y'$, is determined by

$$x' = \frac{a_{21} + a_{22}x + a_{23}y}{a_{11} + a_{12}x + a_{13}y} \qquad\text{(ii)}$$

$$y' = \frac{a_{31} + a_{32}x + a_{33}y}{a_{11} + a_{12}x + a_{13}y} \qquad\text{(iii)}$$

### 3.3.1 Radiometric correction for HiRISE image

For improving the quality of images and reducing the errors in digital number (DN) values, radiometric correction was performed:

$$Z = ([H - B0 * L)]/L) * A0 * G * I * (1.5/AU)2 \qquad\text{(iv)}$$

Where,
H = Output of hiclean (offset and dark current corrected image)
$B_0$ = Intra-channel dark current correction (BIN and TDI dependent)
$A_0$ = Intra-channel gain correction (BIN and TDI dependent)
G = Global gain correction
L = Observation line time
I = I/F conversion factor at Sun-Target distance of 1.5 AU
AU = Distance between Mars and Sun during observation time
Z = Corrected image by radiometrically in I/F units

### 3.3.2 Converting DN to I/F

Intensity flux (intensity/flux, i/f =1) was calculated by converting RDR image DN integer value to flux for getting the normal view of ideal Lambertian reflector to the surface. Intensity/flux values were obtained using the following equation:

$$I/F = (DN * Scaling\ factor) + Offset \qquad\text{(v)}$$



### 3.3.3 Radiometric correction for CTX image

Subtracted and flat fielded CTX row was scaled to radiance using the CTX response coefficient derived during pre-flight calibration:

$$IF = (pix_{val}/exp)/(resp_{coef} * sol\_spect\_rad) \qquad\qquad (vi)$$

Where, $pix_{val}$ is the pixel value being evaluated in DN, exp is the exposure time in m sec, $resp_{coef}$ is the CTX response coefficient in (DN/msec)/(W/m$^2$/µm/sr), and $sol\_spect\_rad$ is the solar spectral radiance for CTX in (W/m$^2$/µm/sr).

<Insert Figure 3. Geomorphological features produced by mosaicking of CTX images >

## 4 Deliberations of result

### 4.1 Fluvial process and landforms

Various landforms of fluvial origin were identified on image although low in aerial extent but important in understanding the evolution of Martin Gale Crater. It was observed that fluvial processes have been active earlier than aeolian processes. The hypotheses of aqueous process were supported by a number of valleys, crater wall, mound and the valley. Spreading down of the valley to the crater floor indicated existence of deep lake in the past crater (Palkey *et al.,* 2004). Each and every feature found in the Martian Gale Crater played their respective roles in the present topography.

### 4.1.1 Canyon

Canyon is an important erosional feature of fluvial process formed during youthful stage of normal fluvial cycle. Canyon is formed by deep cutting of channel and vertical erosion by streams. Significant canyons were identified in the eroded North-West rim of Gale Crater. Rocks sand and silt from crater and from outside also might have been transported by flowing water and deposited on its floor. The details of canyon and the inset HiRISE image show a part of the canyon through a close-up view (Fig. 4 A, B and C).

<Insert Figure 4. (A) View of canyon, (B) CTX image shows the entire canyon and (C) The HiRISE inset image shows the close-up view of canyon >





### 4.1.2 Alluvial fans

Various fan shaped features identified on the surface of crater and provided signs of its formation through lacustrine deposits. Peace valleys fan situated in north western part of Gale Crater has been delineated (Fig. 5 A) and has been shown in CTX image (Fig. 5 B). Irwin *et al.* (2005) hypothesized that highland and valley network might have been reactivated for short period of time and channelled their flow to Gale. This discharge ruined the crater rim leading to lake and delta formation. Similarly, eroded Aeolis mons and crater rim would have been denuded leading to fan formation (Palucis *et al.*, 2014; Anderson and Bell, 2010). Peace vallis fan having concave shape and number of inverted channels were found near western surface of the fan.

<Insert Figure 5. (A) Location of Peace vallis fan and (B) CTX image of Peace vallis fan >

### 4.1.3 Inverted channel

More concentration of lava and eroded coarse grained material was found on fluvial channels than the surrounding topography (Pain *et al.*, 2007; Williams *et al.*, 2009). Relief inversion can be observed through inverted channels formed in the crater and were in large number than the surrounding area (Fig. 6 A to E). These channels must have been active and received enough sand through aeolian process. Low erosion along the channels has resulted in the formation of highlands.

<Insert Figure 6. (A) and (B) shows the inverted channel location. (C) and (D) shows the details of inverted channel through CTX image. (E) The HiRISE inset image of an inverted channel present in the south-west of the Gale Crater >

### 4.1.4 Mound

There is contrasting opinion regarding the accumulation of sediments of mounds through various agents like water, lakes and volcanic eruption, etc. Malin and Edgett (2000) classified Gale as moderately filled impact crater on Mars. They also identified the height of Gale's mound was higher than the crater's rim. Pyroclastic deposits have not been observed as the distance increased from source the surface got thinner. The influence of volcanic activity could not be seen as no ejected material was found. Furthermore, the rocks are found located away from volcanic vent. Hence, deposition of various layers through aeolian work could not be ruled out. Malin and Edgett (2000) also advocated the lacustrine origin of mound by emphasizing thickness and sequential nature of many layered deposits. Wray (2013) advocated that various fluvial and other depositional processes were resulted in mound formation. Fig. 7 (A) and (B) show a view of Aeolis mons upper mound.

<Insert Figure 7. (A) and (B) CTX images shows the peak Aeolis mons mound and the upper mound >





## 4.2 Aeolian process and landforms

The surface of Gale Crater is complex because many processes have been acting on the surface. Palkey *et al.* (2004) observed the pattern of dust cover on the basis of the height of the Gale's mound using thermal inertia, visible and thermal emission imaging system. This analysis indicated that surface of the crater mound was shaped by the action of aeolian process. Ripples, dunes and yardangs landform are the resultant features of such process. Thus, aeolian process found to be a main driver of shaping the surface

### 4.2.1 Ripples, dunes and yardangs

Dunes and ripples in western, north-eastern and near semi-circular shape in southern part of crater were identified. Several features formed by action of wind were found in the impact crater. This is in accordance with the finding of Edgett (2005). The horns of the dunes located in the north-western region of the Gale Crater pointing towards north-eastern direction which indicated blowing of wind from north-western region. However, the morphology of the eastern dunes showed different directions of wind circulation from the north-western region. The circulation indicated that the dunes located in the eastern part might have been formed due to local wind (Sagan and Bagnold, 1975; Carling, 2013; Ami *et al.*, 2016). Fig. 8 clearly shows that the dunes are formed by the material eroded from sedimentary rocks. The dunes are sculpted into yardangs by the salting sand and are present in the eastern and western part of central mound of the Crater (Fig.8 A-E). Similar observations were reported by Malin and Edgett (2000) and Rossi *et al.* (2008) on process and formation of Yardangs.

<Insert Figure 8. (A) and (D) CTX images show the location of dunes and yardangs respectively. (B) and (E) details view of dunes and yardangs. (C) The inset CTX image shows the dunes feature. (F) A close-up view of yardangs through a HiRISE image >

## 4.3 Distribution of ejecta layers

The renewed Martian topography comprises layers or ejecta morphology. These ejecta morphologies can be categorized into single (Fig. 9a), double (Fig. 9b), multiple (Fig. 9c) and radial layers (Fig. 10). The percentage of ejecta was calculated for every 10 latitude on the Martial surface (Fig. 11). Single layer ejecta represents contiguous layer of ejected material having an equatorial diameter of 3 to 20 km and 1 to less than -60 km in higher limits (Hargitai and Barlow, 2014). Maximum distribution of ejecta (8-15%) was found concentrated between -30 and 30 latitudes. The double layer ejecta morphology (nearly 20%) consists of dual faces in which one represents distinctive parallel ridges while outer one is having annular depression located between 20 and 60 latitude (Weiss and Head, 2013).Multiple layer ejecta was embedded with various layers of ejected material having a diameter of up to 90 km (Weiss and Head, 2014). Approximately 16% multiple ejecta layer found between -50 and 40 latitudes. Radial ejecta were more elevated than the neighbourhood terrain (Fig. 10).



Multiple and Radial ejecta layers were found in mid latitudinal Martian topography (Barlow, 2006). The distribution of radial layer ejecta (15-20%) was observed between -40 and 30 latitudes.

<Insert Figure 9. Distribution of various ejecta layers:(a) single layer ejecta,(b) double layer ejecta,(c) multiple layer ejecta >

<Insert Figure 10. Distribution of radial layer ejecta >

<Insert Figure 11. Percentage ejecta at different latitudes >

## 5 Data availability

The Geosciences Node of NASA's Planetary Data System (PDS) archived and distributed digital data related to the study of the surfaces and interior of terrestrial Mars planetary bodies (MODE, 2017). The database presented and described in this article is available at https://ode.rsl.wustl.edu/mars/indexproductsearch.aspx. The PDS Geosciences Node Mars Orbital Data Explorer (ODE) provide search, display and download tools for the PDS science data archives from the Mars Global Surveyor (MGS), the Mars Reconnaissance Orbiter (MRO) and the European Space Agency's Mars Express (ESAME) missions.

## 6 Discussion and Conclusion

The present study has utilized the available visible (MOLA) and topographic (CTX and HiRISE) datasets to explore the geomorphological processes and map the major geomorphic features present on the Gale Crater. The findings revealed that fluvial and aeolian processes have been active in producing various landforms on Martin Gale Crater surface over the ages. Fluvial processes dominated in the earlier stage of geomorphological transformation than the aeolian processes in the crater. The evidence of fluvial process was supported by the presence of various landforms like fluvial channels, inverted channels and fan shaped deposits. Presence of different sedimentary structures and valleys in the crater indicated towards ancient sedimentary deposits due to water action. Aeolian landforms like ripples, dunes and yardangs have also been observed on the surface of Martine Gale Crater. However, inspite of several contrasting opinions regarding the origin of Gale Crater surface, eastern and western part of central mound of the Gale Crater found covered with Yardangs. Multiple episodes of erosion and deposition cannot be ruled out due to erosional unconformities in the mound. Moreover, aeolian deposits were mainly responsible for development of strata observed on Gale and another filled crater. Presence of fan formation and inverted channels in Gale Crater is not sufficient for having possibility of life on mars.

Findings also revealed distribution of varied ejecta layers on the Martin surface. The single layer ejecta was found concentrated between equatorial latitude while double, multiple and radial layers were observed at middle latitudes. These varied topographic features are very much significant in idealizing the surface conditions on Martian topography. Present



study will subsequently ameliorate the bases for further interpretation of the Martian topography. Furthermore, there is need to explore the significant causes held responsible for the deposition of different topographic features on Mars. Studies should be more inclined towards exploring the climate responsible for sediment accumulation, composition of sediments and structure of rocks to reach appropriate evidence of possibility of survival on Martian surface.

**Author contributions:**

HS, JSR, BT designed and proposed the research, PK, SR, OR, KKM processed data, PK, MR implemented techniques, HS, SR, MR analysed results, PK and MR drafted and edited the article. HS and SR joined the discussions and provided constructive suggestions on writing the manuscript.

**Competing interests:** The authors declare that they have no conflict of interest.

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



# List of Tables

**Table 1** Indent of CTX images used for fluvial and aeolian process landforms of canyon, alluvial fan, inverted channel, mound, ripples, dunes and yardangs

| Image ID | Geomorphological features | Canyon | Inverted channel | Peace vallis fan | Peak aeolis mons | Dunes and Yardangs |
|---|---|---|---|---|---|---|
| B03_010639_1752_XI_04S222W.tiff | X | | | | | |
| B21_017786_1746_XN_05S222W.tiff | X | | | X | | |
| F13_040770_1746_XN_05S222W.tiff | X | | | | | |
| G04_019843_1746_XI_05S223W.tiff | X | X | | | | |
| J06_047139_1744_XN_05S221W.tiff | X | | X | | X | X |
| J09_048418_1758_XN_04S222W.tiff | X | | | | | |
| P01_001356_1747_XN_05S221W.tiff | X | | | | | |
| P04_002675_1746_XI_05S222W.tiff | X | | X | | X | |

*(Source:* U.S. Geological Survey's Planetary*)*

**Table 2** Indent HiRISE images identity used for fluvial and aeolian process landforms of canyon, alluvial fan, inverted channel, mound, ripples, dunes and yardangs

| Image ID | Geomorphological features | Canyon | Inverted channel | Peace vallis fan | Peak aeolis mons | Dunes and Yardangs |
|---|---|---|---|---|---|---|
| PSP_001620_1750 | X | | | X | | |
| PSP_001752_1750 | X | | X | | | |
| PSP_002099_1720 | X | X | | | | |
| PSP_001488_1750 | X | | | | X | X |

*(Source: U.S. Geological Survey's Planetary)*

**List of Figures**

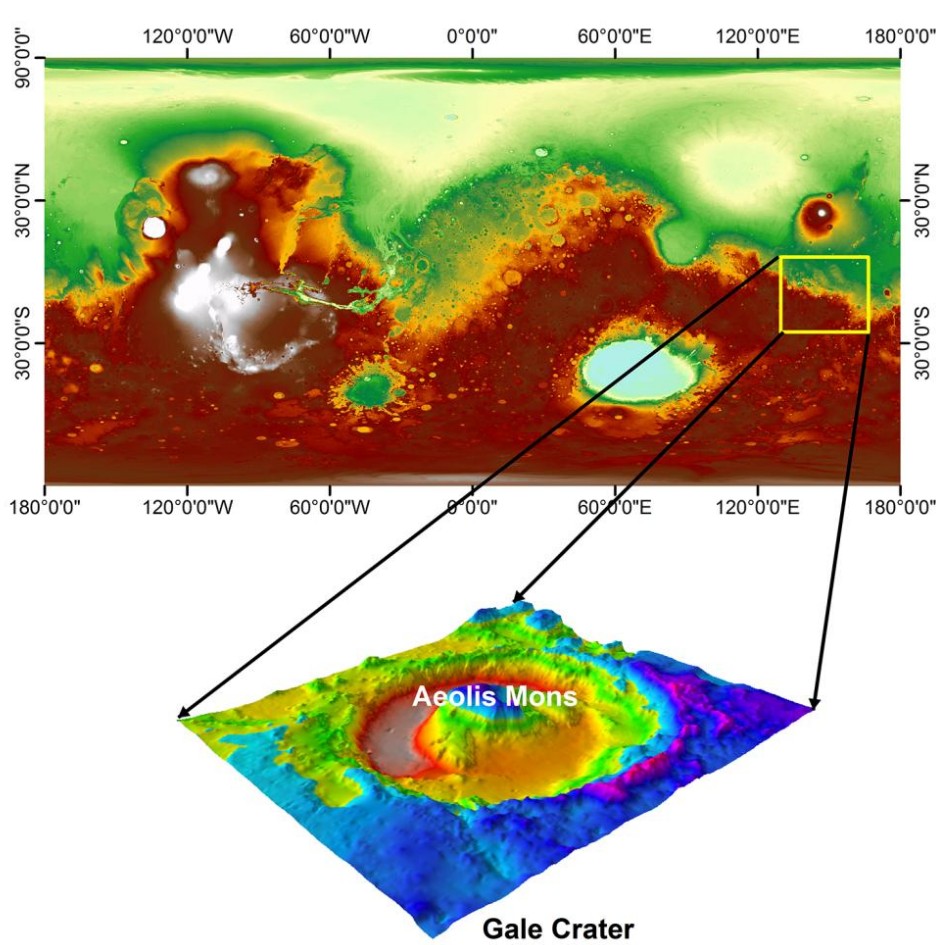

Figure 1. Mars Orbiter Laser Altimeter (MOLA) global topographic map of mars and inset map showing the vertical exaggeration view of Gale Crater and its surrounding



Figure 2. Flowchart showing methodology for mapping and identification of geomorphological features

Figure 3. Geomorphological features produced by mosaicking of CTX images





Figure 4. (A) View of canyon, (B) CTX image shows the entire canyon and (C) The HiRISE inset image shows the close-up view of canyon



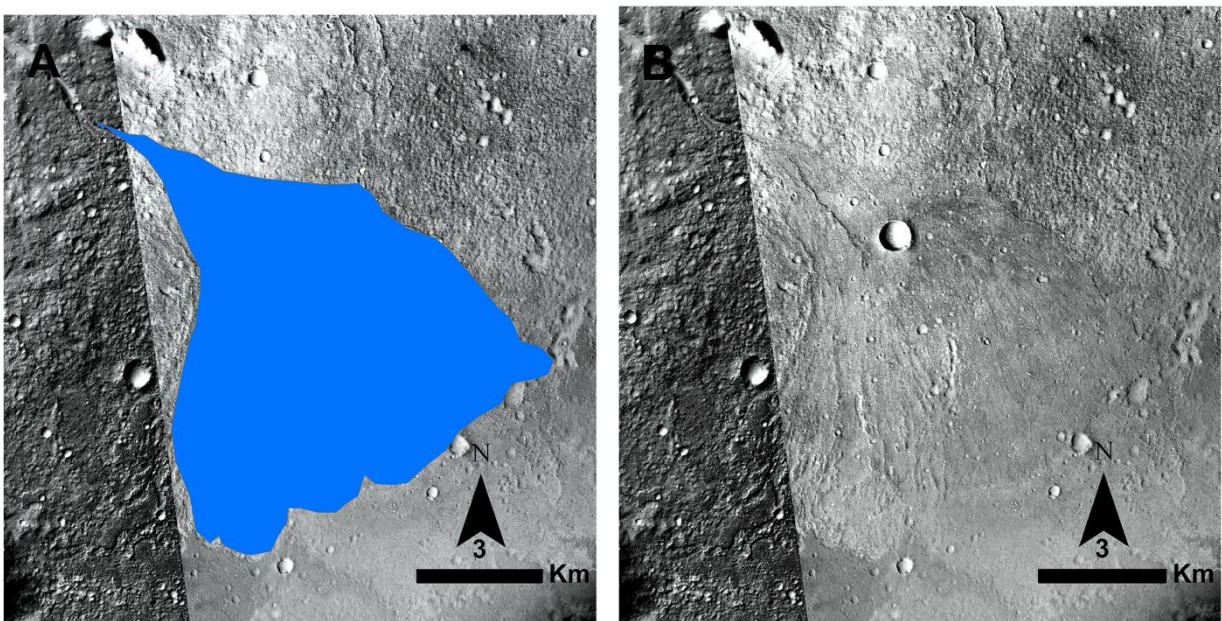

Figure 5. (A) Location of Peace vallis fan and (B) CTX image of Peace vallis fan





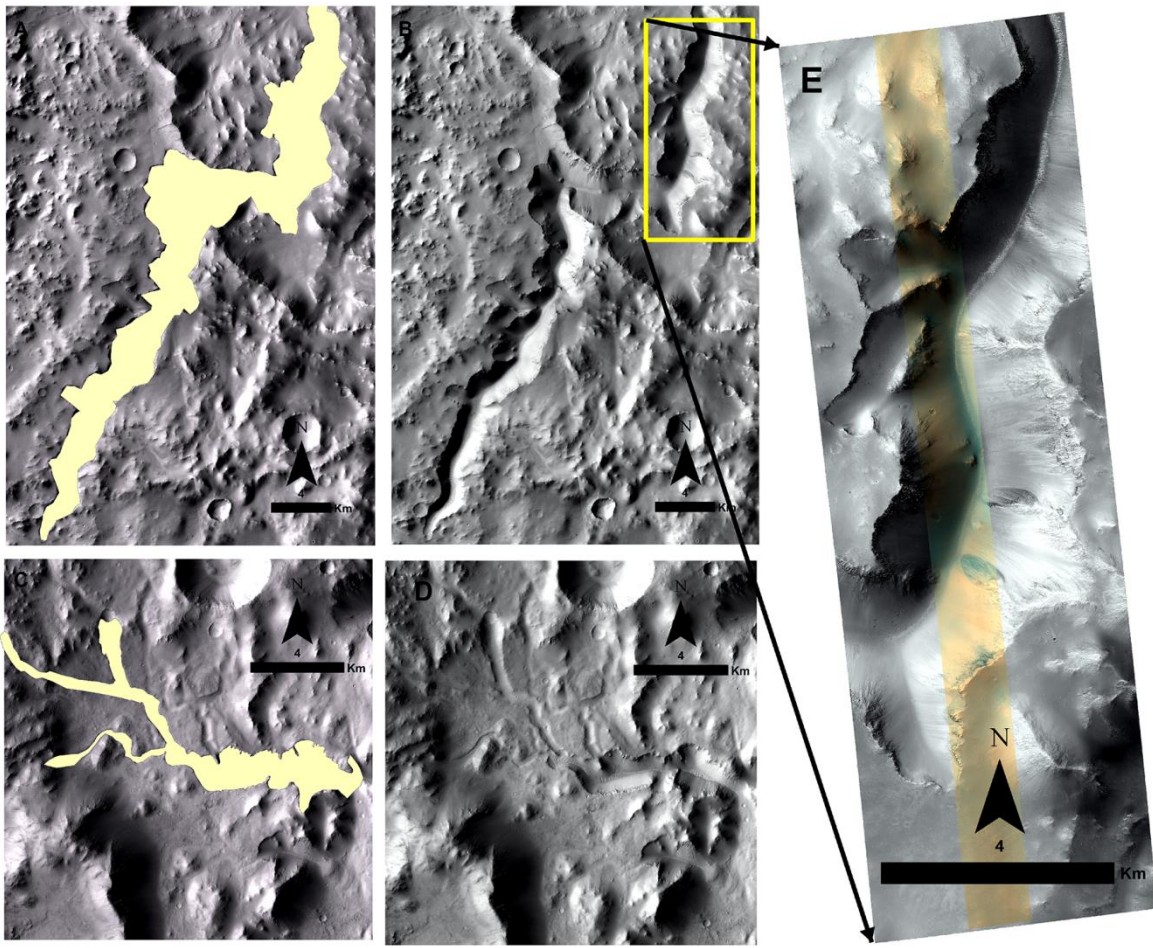

Figure 6. (A) and (B) shows the inverted channel location. (C) and (D) shows the details of inverted channel through CTX image. (E) The HiRISE inset image of an inverted channel present in the south-west of the Gale Crater



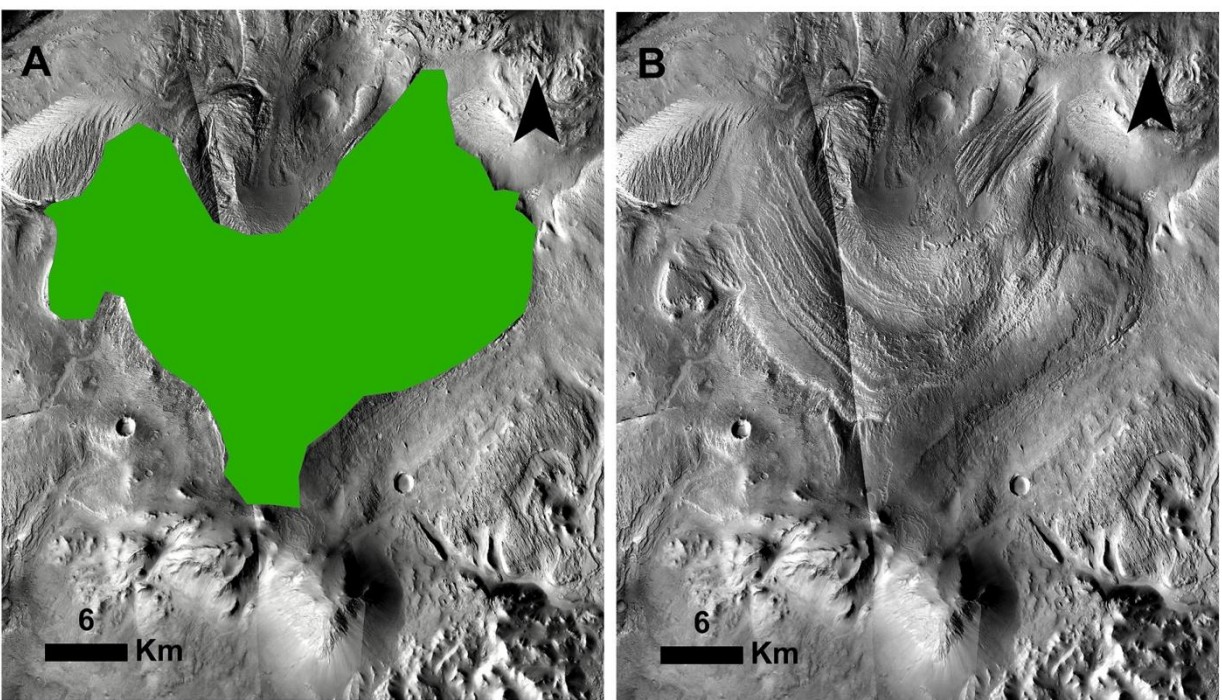

Figure 7. (A) and (B) CTX images shows the peak Aeolis mons mound and the upper mound



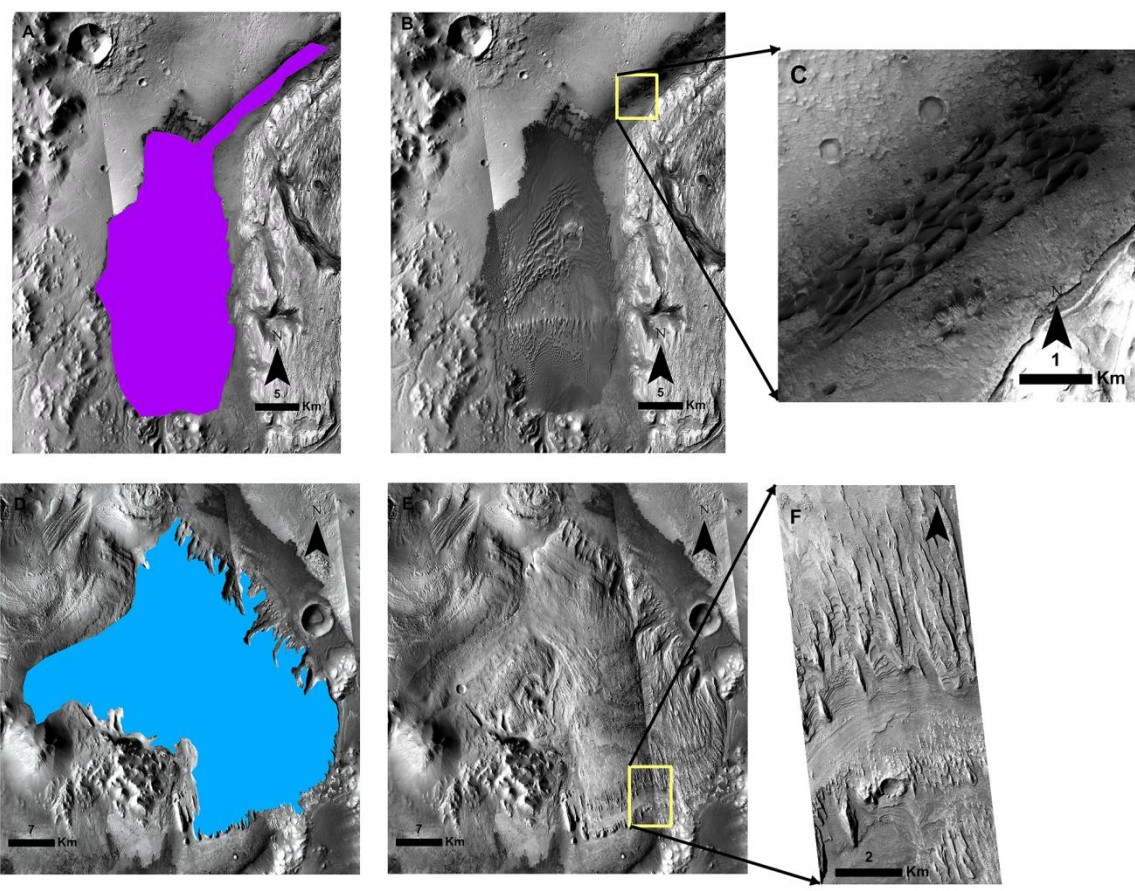

Figure 8. (A) and (D) CTX images shows the location of dunes and yardangs respectively. (B) and (E) detail view of dunes and yardangs. (C) The inset CTX image shows the dunes feature. (F) A close-up view of yardangs through a HiRISE image







Figure 9. Distribution of various ejecta layers: (a) single layer ejecta, (b) double layer ejecta, (c) multiple layer ejecta

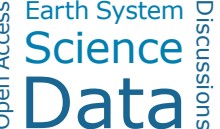

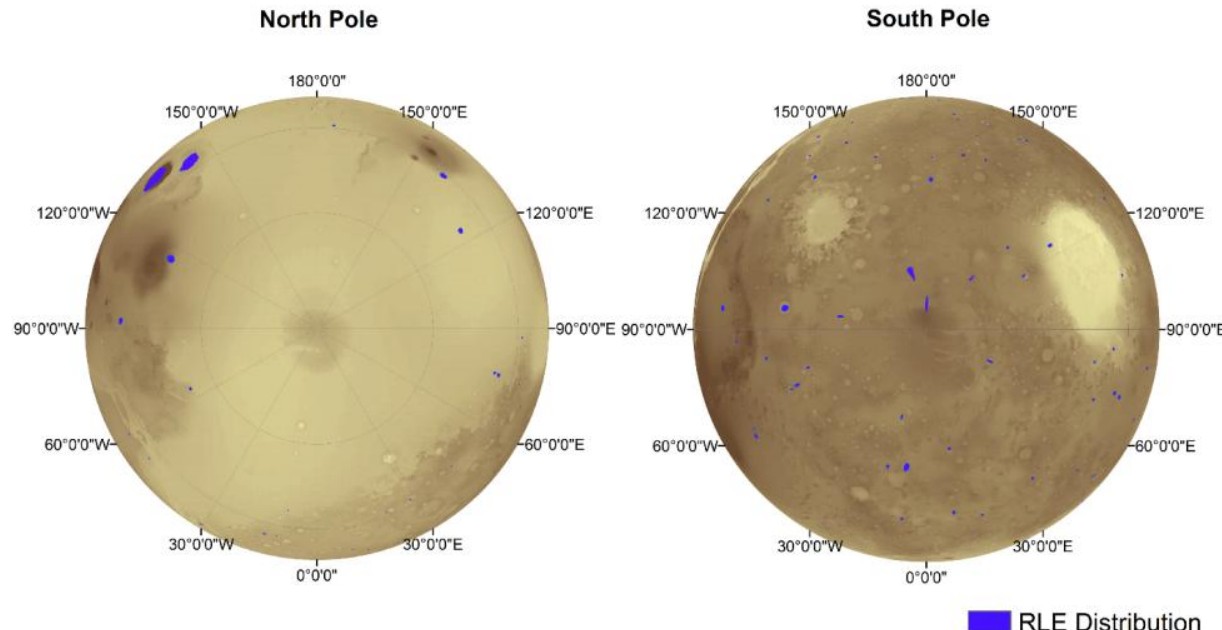

Figure 10. Distribution of radial layer ejecta

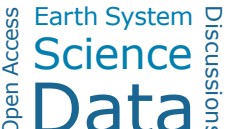


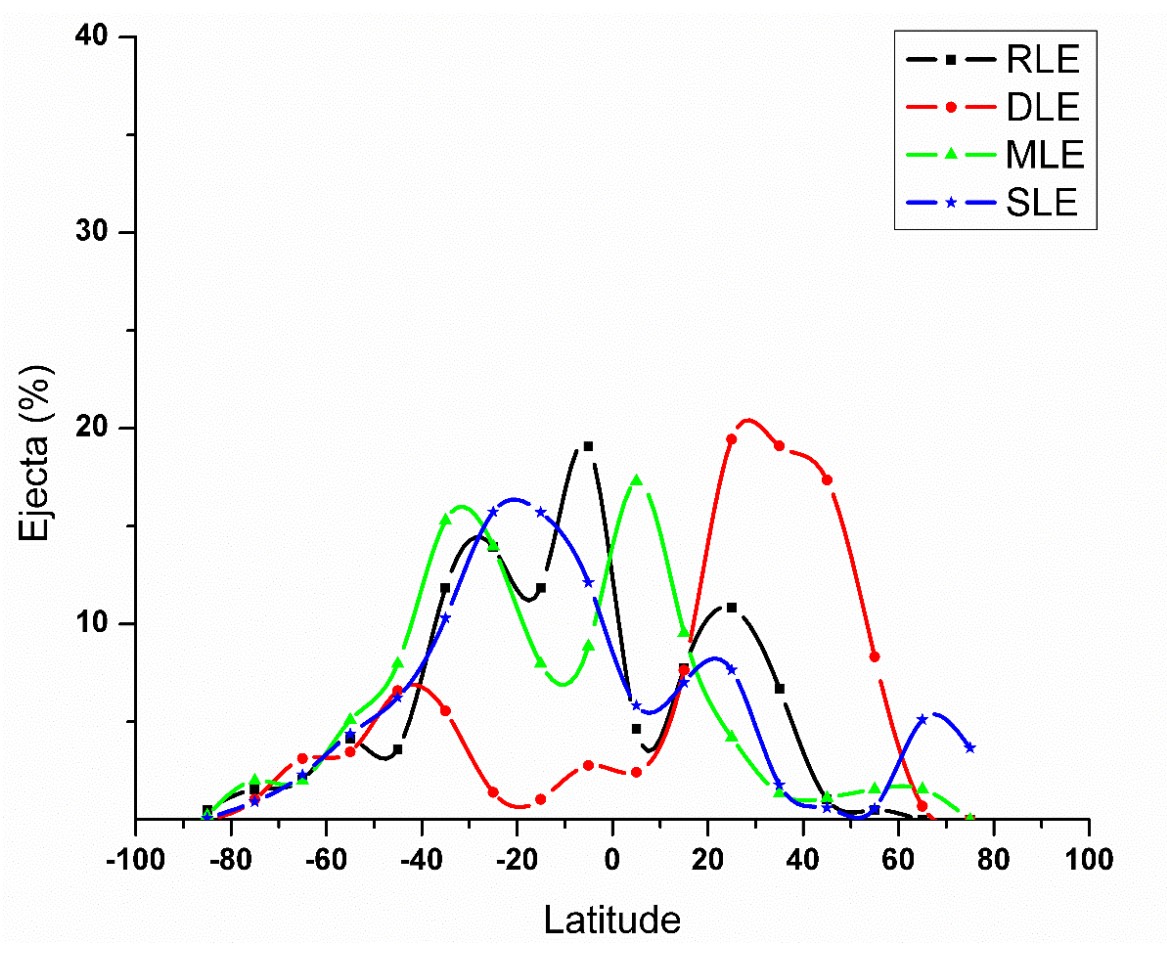

Figure 11. Percentage ejecta at different latitudes