# Peer review of "Exploring Geomorphic Processes and Martian Gale Crater Topography on Mars using CTX and HiRISE Express Image Dataset"

_Earth System Science Data, 2019_

## Short Comment (SC1) · 4 Feb 2019

"mars curiosity rover" should be "Mars Science Laboratory Curiosity rover".
* * *

---

## Referee Comment (RC1) · Anonymous Referee #1 · 5 Feb 2019

This research article presents a geomorphological analysis of fluvial- and wind-related landforms on Gale crater as well as a planet-scale distribution of the different kind of ejecta deposits associated to impact craters.

I have not found a relationship between the geomorphological analysis of the landforms on the Gale crater and the planet-scale distribution of ejecta deposits. I think authors should treat these subjects in different research articles.

The most consistent part of the paper is related to the improvement of the images, where authors seem to have expertise. The poorest parts of the work involve the poorly written English, the explanation about the data and methods used, and the

geomorphological analysis of the landforms. In addition, authors should include more recent references.

It is not clear the topographic dataset which authors have used in this work (MOLA DEM or CTX-derived DEM?). MOLA DEM presents a very-low pixel size (463 m) for performing a precise topographic analysis of the different landforms. In contrast, HRSC- or CTX-derived DEMs (50-75 and 6 m/pixel, respectively) are better for this purpose. On the other hand, the map does not present a frame, geographic grid nor a scale bar. These are key elements of any geologic or geomorphological map, which represents the final result of a geologic/geomorphologic study. The rest of the figures related to the landforms show a low quality that, in my opinion, do not fit the quality criteria of a SCI publication.

---

## Short Comment (SC3) · 5 Feb 2019

mars" should always be written as "Mars". "high lands" should be changed to "highlands", "low lands" changed to "lowlands"

---

## Referee Comment (RC2) · Anonymous Referee #2 · 6 Feb 2019

Review report Journal: ESSD Title: Exploring Geomorphic Processes and Martian Gale Crater Topography on Mars using CTX and HiRISE Express Image Dataset Author(s): Pavan Kumar et al. MS No.: essd-2019-4 MS Type: Research article Special Issue: Linking landscape organisation and hydrological functioning: from hypotheses and observations to concepts, models and understanding (HESS/ESSD inter-journal SI)

This research try to utilize spatially referenced topographic dataset Context Camera images acquired from Mars reconnaissance orbiter for exploring the geomorphic processes and topography of Gale Crater. The method is not novel, but the data is new.

Before publication in the journal of Earth System Science Data (ESSD), there are many part need to improve, I pointed out some doubts, questions and suggestions. I hope it can be useful by author. Finally, I give major revision of this paper.

Title:Too long

1. Introduction

The paper needs to clearly state what are the problems with the existing works (these types of approaches) and what problem(s) this particularly paper was going to address. Without this clearly problem statement readers would have difficulty to see the merit of this paper.

2. Methods The explanation about the data and methods should be improved, I did not find the novelty of the method.

3. Result This part should show some text about validation of your method

4. Discussion

The discussion section is a little poor. There is limited discussion concern the results, and no comparison with previous studies.

5. Conclusion Please give some suggestion and outlook of your research.

Other comment:

Figure: The resolution of figure should be improved.

Reference: The article should be update to the newest.

Language: I am not a native speaker, but I think the paper should be check by a native speaker.

---

## Referee Comment (RC3) · Anonymous Referee #3 · 6 Feb 2019

Exploring Geomorphic Processes and Martian Gale Crater Topography on Mars using CTX and HiRISE Express Image Datasetis a timely and significant contribution in the field of celestial science. Investigation of geomorphological features on Mars has been scantly carried out. The authors have utilized CTX and HiRISE images for mapping geomorphological features. The methodology adopted for this study is sound and results have been presented in convincing manner. The study will immensely be helpful for further studies and add new dimension to existing literature. However, the manuscript is not flawless and needs correction at several places: 1. "mapping of the Mars planet" should be replaced with "mapping of Mars." 2. "The inset vertical exaggeration image provides 3D view of the Gale Crater and its surrounding area" 3D in the sentence

needs to be replaced with perspective in the study area section. 3. "curiosity's science" Curiosity is the US rover currently in operation and its various activities needs to be mentioned in introduction section. 4. "Peace vallis" needs to be replaced with "Peace Vallis". 5. A CTX image is not a topographic dataset. The resolution of CTX imagery is 6 m/pixel. The derived DEM products are typically 18 m/pixel. The manuscript cannot be accepted in its current form and hence demands a minor revision.

―――――――――――――――――

---

## Short Comment (SC4) · 6 Feb 2019

Exploring Geomorphic Processes and Martian Gale Crater Topography on Mars using CTX and HiRISE Express Image Dataset is a timely and significant contribution in the field of celestial science. Investigation of geomorphological features on Mars has been scantly carried out. The authors have utilized CTX and HiRISE images for mapping geomorphological features. The methodology adopted for this study is sound and results have been presented in a convincing manner. The study will immensely be helpful for further studies and add a new dimension to existing literature. However, the manuscript is not flawless and needs correction at several places: 1. "mapping of the Mars planet"

should be replaced with "mapping of Mars." 2. "The inset vertical exaggeration image provides a 3D view of the Gale Crater and its surrounding area" 3D in the sentence needs to be replaced with perspective in the study area section. 3. "curiosity's science" Curiosity is the US rover currently in operation and its various activities need to be mentioned in the introduction section. 4. "Peace Vallis" needs to be replaced with "Peace Vallis". 5. A CTX image is not a topographic dataset. The resolution of CTX imagery is 6 m/pixel. The derived DEM products are typically 18 m/pixel.

The manuscript cannot be accepted in its current form and hence demands a minor revision.

---

## Referee Comment (RC4) · Anonymous Referee #4 · 9 Feb 2019

The present manuscript has utilized the available visible (MOLA) and topographic (CTX and HiRISE) datasets to explore the geomorphological processes on the Gale Crater. The authors found that fluvial and aeolian processes have been active in producing various landforms on Martin Gale Crater surface over the ages. Authors also have explored the major geomorphic features present on the Gale Crater. I recommend the manuscript for publication in Earth System Science Data (ESSD) Journal after minor revision. Some section-wise comments are given below: Introduction: The active mission phase of MSL and some recent literature needs to be added in the introduction section. Study Area: Legend is missing in Figure 1. Without legend how it is possible to interpret the colours and vertical exaggeration. Further, the view presented in figure

[Figure]

1 is not actually 3D rather it is perspective.

Figure 2: Some of the methods shown in flow chart as Crater Counting and JMARS were not discussed in the manuscript.

Material and Methods: Some typographical errors need to be corrected. No references are provided for instruments. How was distortion removed? Orthorectification? Did you use the USGS software ISIS, or a custom code? Results and Discussion: What new knowledge is added to the previous studies?

---

## Short Comment (SC5) · 13 Feb 2019

The authors have presented a good attempt in analyzing the geomorphic processes on the Martian Surface. The paper is written in a lucid way and easy to understand. However, improvement in quality of figures may help in enhancing the overall quality of the manuscript.

---

## Short Comment (SC6) · 14 Feb 2019

Authors have analyzed various landforms of fluvial origin were interpreted on image, although low in aerial extent but important in understanding the evolution of Martin Gale Crater. There should be some correction in manuscript. 1. "Depositional landforms...": The listed landforms should be plural. 2. "mapping of the Mars planet.": Change to just "mapping of Mars."

---

## Short Comment (SC7) · 14 Feb 2019

It has been corrected in main manuscript

---

## Short Comment (SC8) · 14 Feb 2019

Thanks Md Masroor for giving comments. I will take care during final submission

---

## Short Comment (SC9) · 14 Feb 2019

Dear Sayeed Suggestion will be incorporated

---

## Short Comment (SC10) · 14 Feb 2019

Dear Mandal "gale": Change to "Gale" will be incorporate.

---

## Short Comment (SC11) · 14 Feb 2019

Yes Shashikanta. It has been already change. Thanks for giving suggestion to Swati

---

## Short Comment (SC13) · 15 Feb 2019

Dear Vinay Although gale should be Gale in main manuscript but in Table it will remain gale to gale

---

## Short Comment (SC14) · 15 Feb 2019

Dear Author's You have mentioned Mars Orbiter Laser Altimeter (MOLA) data" in the abstract data base but not in the title. I think it should be also in title.

---

## Short Comment (SC15) · 19 Feb 2019

Dear naresh Thanks for constructive comments. I will update it.

———————————————————

---

## Referee Comment (RC5) · Anonymous Referee #5 · 21 Feb 2019

The proposed manuscript intends to investigate geomorphic processes on Gale Crater – Mars using topographic and imagery datasets. I encounter several very important issues: 1) Poor use of the English language; 2) Lack of more recent references; 3) Lack of important references: e.g. Mc Ewen (2007) for HiRISE dataset; Eliason (2009) for radiometric corrections. And many others: MOLA and CTX dataset. 4) The introduction should be divided into: 1) Mars global state of the art view and Gale Crater state of the art. 5) The methodological section is missing some key aspect of the study: how was the image mosaic processed? Which software was used? Standard (ISIS) or developed by the authors? Precision accuracy of processing methodologies? how was Gale crater morphology mapped? 6) How does the presented results differ from the already

presented results of Anderson and Bell (2010)? The presented geomorphological feature map (Fig. 3 and detailed Fig.4, 5, 6, 7 and 8) lacks in detail when comparing it with the (same) map presented in Anderson and Bell (2010). 7) The manuscript title and objectives state geomorphological study of Gale Crater but results and discussion also try to relate eject layers for the global surface of Mars with the morphology of Gale Crater. The methodology does not describe the dataset or methodological approach used for the latter identification. Also, the discussion between the two subjects needs to be improved in order to be of substance in this particular line of investigation. 8) As a science dataset manuscript, I would expect to have access to the results, but authors do not mention where they can be found. No data is presented besides the one we can already encounter at PDS. From all of the above, it is my belief that this manuscript is not yet at the standard of the Earth System Science Data journal and need a thorough revision. Thus, my recommendation is to reject the paper as is.

---

## Referee Comment (RC6) · Adriano Ribolini (Referee) · 23 Feb 2019

General comment

The topic of the paper is surely interesting and stimulating. I am not sure if the results provide evidence really new for the specific area and relevant for the knowledge of the shaping of the Martian surface. In case, the Authors should stress the novelty and the relevance of their results. Geomorphologic and geologic terminology in some points is inadequate. Description of erosive and depositional landforms is minimal, and mutual relations between surface features of different origin should have been more described. English is not my native language, but along the text I found many

points hard to be understood. Here below some specific comments, while others are reported as annotations to the PDF file of the manuscripts (attached)

Introduction What seems problematic is the apparent mix of results referred both to the Mars' surface in general and the Gale crater. Moreover, some references are lacking. In general, the relevance of the paper's goal is not stressed enough Materials and Methodology There are some sentences that belong to the results section (see Crater morphology identification). Indeed, figure 3 should be better placed after the illustration of every mapped landforms, Results There are many unclear sentences, affirmations that needs to be supported by evidence, and some descriptions incomplete or lacking. I understand that the origin of mounds is complex, but I was expected to find an authors' opinion even prudent and not conclusive. About the Aeolian process and landforms, it is not so evident by the figure 8 that the material forming the dune is coming from eroded sedimentary rock. Please provide a better explanation. The result section 4.3 Distribution of ejecta layers appears disconnected from the motivation of the paper, because, despite a distribution extended to the total Mars surface, no evidence are presented in the study area. I suggest to eliminate this part, or to add some images and descriptions coming form Gale Crater

Conclusion Conversely to what reported, clear evidence of mutual relations between fluvial and aeolian features suggesting a temporal sequence are not presented in the paper. It would have been interesting to see an image where fluvial landforms are overlapped by aeolian sediments. The 4.2.1 paragraph says that dunes are formed by material eroded by sedimentary rocks (fluvially eroded? however not so evident by the figure 8), and then shaped by wind action in yardangs. It is not clear if fluvial processes were dominating the earlier stage of surface shaping. I think that there is a confusion between sedimentary structures and depositional landforms, which are not synonyms.

Best wishes

Please also note the supplement to this comment:

https://www.earth-syst-sci-data-discuss.net/essd-2019-4/essd-2019-4-RC6-supplement.pdf

[Figure]

**Supplement:**

[revised manuscript text omitted]

---

## Editor Comment (EC1) · Giulio G.R. Iovine (Editor) · 3 Mar 2019

Allegations of plagiarism on manuscript ESSD-2019-4

The editors of ESSD received an email containing allegations of plagiarism on the manuscript ESSD-2019-4, entitled: "Exploring Geomorphic Processes and Martian Gale Crater Topography on Mars using CTX and HiRISE Express Image Dataset", by Pavan Kumar and co-authors. The content of such email is included below. Authors and reviewers are strongly urged to pay attention to the content of such email, and to act accordingly: - the authors, to either provide exhaustive explanations or withdraw their manuscript; - the reviewers, to verify the actual "similarities" in the contents

between the manuscript ESSD-2019-4 and the paper published in 2010 by Ryan B. Anderson and James F. Bell III (Geologic mapping and characterization of Gale Crater and implications for its potential as a Mars Science Laboratory landing site, Mars, doi: 10.1555/mars.2010.0004)

Please, note that - as all the manuscript received for publication in ESSD - even ESSD-2019-4 had automatically been checked to verify any plagiarism problem before starting the public discussion/review phase. The manuscript ESSD-2019-4 passed the test successfully: in fact, the iThenticate.com Similarity Report (completed on 18 Jan 2019), provided by Copernicus, stated that similarities were "negligible / not found".

After discussing among the Editors and contacting the author of the email (who prefers to remain anonymous), I'm now posting the content of the email as the Handling Topical Editor to ensure editorial transparency and to allow for an adequate discussion on this matter. Many thanks to all for your attention and precious support. Giulio Iovine Handling Topical Editor ESSD

*

The text of the received email follows:

<Dear Editors,

With due respect, I have a doubt regarding the novelty and originality of the manuscript entitled: 'Exploring Geomorphic Processes and Martian Gale Crater Topography on Mars using CTX and HiRISE Express Image Dataset' with doi: 10.5194/essd-2019-4.

Most of the findings in the paper seems a duplicate version of the paper published in the journal " Mars" in 2010 by Ryan B. Anderson and James F. Bell III. Title of the paper: 'Geologic mapping and characterization of Gale Crater and implications for its potential as a Mars Science Laboratory landing site' doi: doi:10.1555/mars.2010.0004

Although the authors [Kumar et al.; correspondence: Meenu Rani (meenurani06@gmail.com)] mentioned the reference of this paper, but I believe they have

not not justified the use of the original work of Anderson and Bell (2010) and used it inappropriately. Only the mode of representation is a bit different with different colour schemes and views used in the figures of the ESSD paper, with similar findings as of the 'Mars' paper; the language being modified and without any result validation. However, the language of the ESSD paper is erroneous too. Findings from other papers are also included, but mostly from Anderson and Bell (2010). I am enclosing both the manuscripts for your ready reference. Kindly check as I might be wrong too, but I have full faith in your intelligence and decisions. The rest I leave it to your wisdom and your fine sense of judgment.

Regards.>

---

## Author Comment (AC1) · 4 Mar 2019

Doubt of anonymous author of mail regarding the novelty and originality of the manuscript:

The study provides the first analysis of the distribution of craters with displaying MLE morphologies. The study finds that MLE craters concentrate along the dichotomy boundary and in areas displaying channels such as those near Elysium and Tharsis.

Anonymous author says that most of the findings in the paper seems a duplicate version of the paper by Anderson and Bell (2010):

[Figure]

The location of geomorphological features of an area will be the same but in case of Mars it is necessary to find out the mode of origin. Various authors presented their views and reviews towards the same features in different way since the origin of the Gale Crater formation cannot be changed but its evidence can be explained in different ways. Single layer ejecta (SLE) is the most common ejecta morphology over the entire Martian surface. Martian impact craters display a variety of ejecta and interior features differing from those seen on dry and atmosphere free bodies like the moon.

Anderson and Bell (2010) were examined the geological and geomorphic characteristics of Gale crater to identify the potential landing site for MSL.

In our study the geomorphic features mainly of aeolian and fluvial origin were identified and delineated.

Below are the points to discard the duplication of Anderson and Bell version:

1. Anderson and Bell (2010) carried out their research about geologic mapping and characterization of Gale Crater using mathematical explanation and earlier published work.

2. We used the information contained in the Barlow catalogue of Large Martian Impact Craters to discuss the major geomorphic features and chronological description of fluvial and aeolian processes. The Catalogue contains information on 42,284 impact craters >= 1.7 km diameter distributed across the entire surface of Mars. The catalogue was compiled from digitization of the Viking 1:2,000,000-scale photo mosaic produced by the U.S. Geological Survey in the 1980s (Barlow 1987). Catalogue 1.0 can be accessed through the U.S. Geological Survey's Planetary Interactive GIS on the Web Analyzable Database (PIGWAD) (http://webgis.wr.usgs.gov).We also identified significant ejecta morphologies on Martian surface.

3. Anderson and Bell (2010) used visible (CTX, HiRISE, MOC), infrared (THEMIS, CRISM, OMEGA) and topographic (MOLA, HRSC, CTX) datasets and data products to
conduct a study of Gale Crater, with a particular focus on the region near the proposed Mars Science Laboratory (MSL) landing site and traverse.

4. We used only visible (CTX, HiRISE) and topographic (MOLA) datasets to explore the Martian surface and the crater deposition on Northern site of Mars with Single Ejecta.

5. Anderson and Bell (2010) explained fluvial and aeolian weathering through mathematical explanation and earlier studies.

6. In our study geomorphic features were identified using the visual interpretation and catalogue information. It has also been suggested thatafter the impact of a meteorite the soils got loosen and susceptible to both fluvial and aeolian weathering. No duplication in the findings of fluvial processes and landforms rather it was in accordance with Pelkey et al. (2004).

7. Anderson and Bell (2010) studied a detailed geomorphic feature on the Gale Crater for possible MSL landing site.

8. While our findings revealed that significant canyons were identified in the eroded north western rim of theGale crater. Peace valley fan situated in north western part of the Gale crater were delineated in our study. These fans were formed due to denudation of Aeolis mons and crater rim were also reported by Palucis et al. (2014), Anderson and Bell (2010). These have been cited in the manuscript.Our study revealed that inverted channels must have been active and received enough sand through aeolian process. Low erosion along the channels has resulted in the formation of highlands.

9. Anderson and Bell (2010) were unable to rule out a lacustrine or aeolian origin for the lower mound using presently available data.

10. Our study revealed deposition of various layers to aeolian work. Malin and Edgett (2000) also advocated lacustrine origin of the mound by emphasizing thickness and sequential nature of many layer deposits. Our analysis indicated that the surface of the crater mound shaped by the action of the aeolian process and it is the main driver

for shaping the surface. In our study dunes are formed by erosion of sedimentary rockswhile Anderson and Bell (2010) suggested the basaltic composition of the dunes.

11. There was no mention of Ejecta morphologies in the work of Anderson and Bell (2010).

12. We have provided separate figures (Figure. 9,10 and 11) and a paragraph for the ejecta morphologies in the manuscript.

How our work is different from Anderson and Bell (2010):

a) We have used limited data and different methodologies.

b) Java Mission-planning and Analysis for Remote Sensing (JMARS)was used in our study to generate the spatial datasets as a CTX image stamp map.The required images for Gale crater were identified in JMARS and acquired from the Mars Orbital Data Explorer website.

c) We have used Barlow crater catalogue (Barlow, 2003).

d) Origin of geomorphological features were correlated with ejecta and has been provided in the supplementary file.

e) Relationship was established between Crater diameter and pit diameter in single layer ejecta for the formation of the mound.

f) Our study is confined to aeolian and fluvial features only.

Dissimilarities in Figures:

a) Anderson and Bell (2010) in the Figure 7 have shown various geomorphic features on the Gale Crater.

b) While in our manuscript the Figure 3 is representing only the fluvial and aeolian features identified on the Gale Crater.

c) All the Figures from 4 to 9 in our manuscript are different from the Figures of Anderson and Bell (2010).

d) In our study we suggested the impact of ejecta in the formation of the fluvial and aeolian features on the Gale crater. Our study suggested that meteorite effect may have led to the origin of fluvial and aeolian process on the Gale crater.

e) Anderson and Bell (2010) usedmathematical explanation for explaining the identified geomorphic features on Martian Gale Crater.

f) While in our manuscript Barlow Crater catalogue (2003) was used containing information of all the Martian craters.

---

## Author Comment (AC2) · 4 Mar 2019

Dear Sir Thank you for providing constructive comments. I appreciate your effort for improvement of this manuscript.

Response 1. This research article presents a geomorphological analysis of fluvial- and wind-related landforms on Gale crater as well as a planet-scale distribution of the different kind of ejecta deposits associated to impact craters. I have not found a relationship between the geomorphological analysis of the landforms on the Gale crater and the planet-scale distribution of ejecta deposits. I think authors should treat these subjects in different research articles. Authors Response: Planet-scale distribution of

ejecta deposits has been correlated with geomorphological analysis because it is most important part of this research.

2. The most consistent part of the paper is related to the improvement of the images, where authors seem to have expertise. Authors Response: All the images have been re-modified according to standard of ESSD policy.

3. The poorest parts of the work involve the poorly written English, the explanation about the data and methods used, and the geomorphological analysis of the landforms. Authors Response: All literature has been carefully re-written.

4. In addition, authors should include more recent references. Authors Response: Most recent reference has been added in literature part of manuscript.

5. It is not clear the topographic dataset which authors have used in this work (MOLA DEM or CTX-derived DEM?). MOLA DEM presents a very-low pixel size (463 m) for performing a precise topographic analysis of the different landforms. In contrast, HRSC- or CTX-derived DEMs (50-75 and 6 m/pixel, respectively) are better for this purpose. Authors Response: Authors have used CTX-derived DEMs data set.

7. On the other hand, the map does not present a frame, geographic grid nor a scale bar. These are key elements of any geologic or geomorphological map, which represents the final result of a geologic/geomorphologic study. Authors Response: A frame, geographic grid nor a scale bar has been added in main Figure 3 and also added in all produced images.

8. The rest of the figures related to the landforms show a low quality that, in my opinion, do not fit the quality criteria of a SCI publication. Authors Response: All the images have been re-modified at 500 DPI according to standard of ESSD policy.

---

## Referee Comment (RC7) · Anonymous Referee #4 · 11 Mar 2019

Dear colleagues,

Thank you for your mail. I have already submitted my review on the manuscript entitled "Exploring Geomorphic Processes and Martian Gale CraterTopography on Mars using CTX and HiRISE Express Image Dataset". I read another manuscript mentioned in your mail by Anderson and Bell to verify the actual similarities between the two. While comparing the manuscripts, I found that author did not study both manuscripts seriously. The detailed comparison of the manuscripts is discussed here:

First of all, the objectives of the both papers are different from each other. Anderson and Bell (2010) focused on the geologic characteristics of the Gale crater and poten-

tial landing site for Mars Science Laboratory landing. Therefore, they studied surface features on the crater for probable landing site. However, Kumar et al. primarily focused on the surface features and their geomorphic processes on the Gale Crater. They used their knowledge and understanding of surface processes for identification and delineation of geomorphic features present in the crater.

After comparing the results and findings of the both papers, it is clear to me that both have different dimensions and thoughts. Anderson and Bell (2010) analysed their results based on the mathematical findings. Their analysis was based on the numerical values of elevation, length, CTX albedo and Thermal Inertia. With the help of these factors, morphology of the crater topography was studied for potential landing site. Their prime objective was to identify a location which is fit for four primary criteria. This includes diversity, context, habitability and preservation potential. Their study found suitable landing site for MSL where rover could explore numerous geomorphic features. The diversity of the topographical features, geological and chemical evidence for the past for habitability are of paramount importance to their study for site selection.

Kumar et al. studied the surface processes and identified landform formation based on their knowledge on geomorphic processes and some earlier studies. Their results and findings are theoretical in nature. Mars is a celestial body where in situ data for result validation is not possible for every researcher. Therefore, based on the earlier study and the knowledge from the evidence of the earth's surface processes were used here. They have studied about the distribution of ejecta layer formation based on the spatial distribution of different latitudes. This formation is based on the theory of asteroid impact where Anderson and Bell (2010) focused on the volcanism for their study. In this way both manuscripts have different dimensions from each other.

Anonymous author of the mail may be confused by the Figure 3 of Kumar et al. with Figure 7 of Anderson and Bell (2010). There are some similarities and dissimilarities in the figure and have reasons:
Similarities: Dark toned layered yardangs, dark toned dunes and upper mound of Anderson and Bell (2010) have similarities with Kumar et al. Figure 3. These units are very large in Gale Crater which can be easily identified from Mars obiter image. Therefore, it is not an issue as per my understanding.

Dissimilarities: Various small features were mapped by Anderson and Bell (2010) which are absent from Figure 3 of Kumar et al. Anderson and Bell (2010) mapped every detail from inside and outside of the crater for their possible landing site of MSL. However, Kumar et al. focused on the inside geomorphic features which makes it different form Anderson and Bell (2010).

Kumar et al. cites Anderson and Bell (2010) many times in their paper as an evident of earlier researches in Gale Crater. The work by Anderson and Bell is an extensive study on different aspects and is different from the manuscript by Kumar et al. The anonymous person should come forward for interactive discussion as the other scientific community is participating. If anybody has doubt and belief of any research related query, it can be confirmed either from the authors or through open discussion platform.

My submission is that the authenticity of the anonymous person should be ascertained and checked.

I think that the manuscript submitted by Kumar et al. be accepted for publication.
* * *

---

## Editor Comment (EC2) · Giulio G.R. Iovine (Editor) · 20 Mar 2019

Based on the trail of reviews, comments, and responses, the manuscript ESSD-2019-4 suffers from serious flaws, both in scientific content and form. The Journal ESSD is devoted to the publication of new data concerning the planet Earth, rather than data interpretation. In this particular case, the discussed data even refer to another planetary body. The manuscript does not provide details on how to retrieve data from the NASA repository. The authors processed publicly available data, but neither produced nor shared an improved derived dataset of their own. The results of reviews recommended outright rejection in two of six cases during the public discussion phase. In addition,

the manuscript has been criticized by the Mars research community. ESSD serves the earth system research community with high-quality data with potential for reuse. This manuscript fails to provide it and gives users neither hint nor guidance to the validity of the product. According to the above considerations, the editorial team recommends rejecting this manuscript.